# Imaging in Diagnosis of Systemic Sclerosis

**DOI:** 10.3390/jcm10020248

**Published:** 2021-01-12

**Authors:** Katarzyna Rutka, Adam Garkowski, Katarzyna Karaszewska, Urszula Łebkowska

**Affiliations:** 1Department of Radiology, Medical University of Białystok, M. Skłodowskiej-Curie 24A, 15-276 Białystok, Podlaskie, Poland; katarzyna.rutka@op.pl (K.R.); ursleb@o2.pl (U.Ł.); 2Department of Rheumatology and Internal Medicine, Medical University of Białystok, 15-276 Białystok, Podlaskie, Poland; katarzynaszp21@gmail.com

**Keywords:** systemic sclerosis, imaging, X-ray, ultrasound, computed tomography, magnetic resonance imaging

## Abstract

Systemic sclerosis (SSc) is a connective tissue disease characterized by fibrosis in skin and internal organs, progressive vascular obliteration, and the production of autoantibodies. Diagnostic imaging is irreplaceable in both diagnosing and monitoring patients suffering from systemic sclerosis. In addition to routinely used methods, such as comparative X-ray of the hands or a contrast-enhanced examination of the upper gastrointestinal tract or chest, there is an array of less widespread examinations, with an emphasis on magnetic resonance imaging (MRI) and ultrasonography, not only in the evaluation of the musculoskeletal system. This article will review the various imaging modalities available for SSc imaging and assessment, focusing on their utility as tissue-specific diagnosis and treatment monitoring.

## 1. Introduction

Systemic sclerosis (scleroderma, SSc) belongs to a wide spectrum of connective tissue diseases—a diverse group of disorders characterised by chronic inflammation of connective tissue, usually of autoimmune origin [1]. The most distinctive features of SSc include small blood vessels abnormalities, autoantibodies’ production, and fibrosis. Symptoms might involve any organ systems—the most common manifestations involve the skin, lungs, gastrointestinal (GI) tract, musculoskeletal system, and urinary tract [2].

SSc typically affects women (4:1 predilection), with the median onset at the age of 30–50 years. There is a higher morbidity within black patients than in other ethnical groups. A very high prevalence of SSc in Choctaw Native Americans suggests the possibility of a genetic factor in the pathogenesis of this disease. Reports of familial history of SSc or other autoimmune disorders in SSc patients have also shown support for the genetic grounds of SSc. The incidence rate seems stable over several decades in both Europe and the United States [1,2,3,4,5]. The efficacy of modern treatment allowed to improve the survival rate of SSc patients. Renal insufficiency used to be the leading cause of mortality in SSc patients, but is presently successfully treated. The most common cause of morbidity is currently lung involvement—pulmonary hypertension (PH) and/or pulmonary fibrosis [6].

The diagnostic criteria for SSc are still not universal. The classification criteria of the American College of Rheumatology (ACR), presented in 1980 and updated by a collaboration of ACR and European League Against Rheumatism (EULAR) in 2013, are used most commonly. According to those criteria, skin thickening proximal to the metacarpophalangeal joints is sufficient proof of SSc. If it is not present, seven other signs should be evaluated: (1) skin thickening of the fingers (either sclerodactyly or puffy fingers); (2) fingertip lesions (digital tip ulcers or fingertip pitting scars); (3) telangiectasia; (4) abnormal nailfold capillaries; (5) pulmonary arterial hypertension (PAH) and/or interstitial lung disease; (6) Raynaud’s phenomenon; and (7) SSc-related antibodies: anticentromere and/or anti-topoisomerase I (anti-Scl-70) and/or RNA polymerase III [7,8].

The criteria listed above have varied scores that are used to calculate an overall score; patients whose total score exceeds 9 are classified as having definite SSc. The specificity of the 2013 criteria is estimated at 90% and their sensitivity at 91%, which is a valid improvement compared with the 1980 recommendations (of specificity and sensitivity of 75 and 72%, respectively [7,8].

Some of the evaluated symptoms are only seen in patients with an advanced disease, which was emphasised by Medsger et al.—who proposed diagnostic criteria for the early stage of scleroderma. Raynaud’s phenomenon is the most frequent symptom of the early stage [3]. However, while it is seen in the majority of SSc patients, it is not specific—only one tenth of patients with Raynaud’s phenomenon would develop systemic SSc [9].

Two basic forms of SSc are distinguished: limited cutaneous scleroderma (lcSSc) and diffuse cutaneous scleroderma (dsSSc). The limited form of the disease is more common—seen in 80% of patients [10]. The diffuse form is associated with a significantly more expansive systemic involvement, including GI tract, heart, and lungs. LcSSc is frequently associated with a distinctive combination of symptoms, previously known as the acronym CREST syndrome—including soft tissue calcifications, Raynaud’s phenomenon, esophageal dysfunction, sclerodactyly, and telangiectasias [11]. However, the prevalence of both pulmonary fibrosis and pulmonary arterial hypertension is similar in the limited and diffuse form [12].

Modern medical imaging methods are widely applied both in diagnosing of SSc and in further management in patients suffering from scleroderma. Summary of indications and pitfalls for various imaging modalities in various clinical forms of systemic sclerosis is summarized in Appendix A.

## 2. Classical Radiology

### 2.1. Musculoskeletal System

Hands are the most characteristic location of lesions in SSc. Articular symptoms are seen in 10–60% of patients at the time of diagnosis. Features allowing to distinguish changes caused by scleroderma from symptoms of other diseases are predominantly acro-osteolysis, calcifications, and soft-tissue thinning, which are all visible on radiographs [13]. According to the latest research, acro-osteolysis (Figure 1 and Figure 2) is seen in about 16% of patients [14]. The distinctive ‘sharpened pencil’ sign is caused by resorption of the terminal tuft of distal phalanges; this symptom is usually more pronounced on the palmar surface of phalanges. A correlation between acro-osteolysis and digital ulcerations was observed. The less common locations where bone resorption might be seen in SSc patients include medial parts of the ribs, distal end of the radial and ulnar bones, clavicle, and mandibulum [15,16]. Although acro-osteolysis is a common finding in SSc, a number of other pathological entities might be also associated with this symptom—including rheumatological disorders, psoriatic arthritis, hyperparathyroidism, and thermal injuries [17,18].

The soft-tissue thinning is visible and may be objectively evaluated on a hand radiograph—by comparing the thickness of tissue of the fingertip and at the basis of the distal phalanx. A ratio below 0.2 is an unambiguous symptom of peripheral soft tissue thinning. Moreover, repeated calculation of this parameter might be useful in the monitoring of the disease progression [19].

Soft-tissue calcifications are observed in about 46% of patients [20]. Both subcutaneous and periarticular locations are possible. Periarticular calcifications were reported not only in typical peripheral regions (e.g., in the hand), but also in less frequent areas, such as the paraspinal site, including even the spinal canal itself, with a possibility of its narrowing [15].

However, joint symptoms are frequent in SSc, they are rather a manifestation of periarticular lesions than an actual arthritis. Nevertheless, scleroderma patients’ radiographs often demonstrate features that are typical for arthritis—for example, periarticular osteopenia and joint space narrowing—seen in a varying percent of SSc patients (9–54%) [21,22,23,24]. The specific location of the erosions might be useful in differential diagnosis of rheumatioid arthritis and SSc—in sclerodermam lesions are usually seen in the distal interphalangeal joints and the dorsal portion of metacarpophalangeal joints. The distal interphalangeal locations is unfortunately also frequent in psoriatic arthritis. The sclerodermic erosions are generally small, especially when compared with the rheumatioid arthritis patients. The joint lesions are classified as minimal changes (in 20% of patients), periarticular fibrosis (30–34%), degenerative lesions (13–22%), or inflammatory lesions (13%) [21,25].

Flexion contractures are present in approximately one-third of patients. Their most common sites are the metacarpophalangeal joints and interphalangeal joints. They are more common in diffuse scleroderma [25].

### 2.2. Respiratory Tract

Pulmonary fibrosis (Figure 3) might be observed on a chest X-ray only in patients in an advanced stage, thus largely diminishing the role of classical radiology in the diagnosis of pulmonary involvement in SSc patients [21].

The pulmonary arteriography (right heart catherization) remains the gold standard in the evaluation of PAH [26]. Right heart catherization is required to confirm the diagnosis of PAH and to assess the severity of haemodynamic impairment [27].

### 2.3. Gastrointestinal Tract

Symptoms involving the GI tract are the second most common in SSc patients, preceded only by skin lesions. Complaints are reported in approximately 50% of patients [28]. Esophageal location is more typical to limited scleroderma—seen in as many as 90% of patients, but all of the other gastrointestinal symptoms are more prevalent in the diffuse form of the disease [29].

Videofluorography swallow study of the esophagus or of the whole upper GI tract allows to evaluate the motor function of the esophageal motility. Typically altered peristalsis is observed in the distal esophagus, but in advanced phase, it might involve the proximal third as well [30]. Usually patients show a decrease of the pressure of the lower esophageal sphincter (Appendix A). Gradually the lumen of the esophagus becomes extended. These symptoms are easier to be visualised in horizontal position, so the examination should be conducted in both supine and prone positions [31]. The results of a contrast enhanced examination are corresponding with the values obtained by a manometric examination (convergence of approximately 60–70%) [32]. It is useful to remember, that administration of barium sulfate might exacerbate small bowel related symptoms, due to abnormal peristalsis and suscebility to constipation—thus iodine swallow should be considered a viable alternative [16]. Some of the patients additionally demonstrate esophageal or gastro-esophageal reflux, which might be also visualised in an contrast-enhanced study. A iodine or barium swallow remains a useful tool in evaluation of esophageal involvement in SSc patients [33].

Stomach involvement is observed in 10–75% of patients. The prevalence of of gastric electrical activity dysfunction with cutaneous electrogastrography (EGG) is the most common. In advanced cases, a delayed voiding of the stomach might be noted in scintigraphy or radiography using radiopaque pellets. Watermelon stomach (Gastric Antral Vascular Ectasia, GAVE) is a less common complication of SSc, exhibited by up to 5.7% of patients [26,34,35]. 

### 2.4. Oral Cavity

A panorex examination allows to show homogenous widening of the periodontal space in one tenth of the patients [31]. In twice as many, bone resorption is reported—located within the mandibular condyle, its coronoid process, or rarely in the mandibular angle [36].

## 3. Ultrasound

### 3.1. Skin

The integumentary system is the most common location of systemic lesions in scleroderma. Thickening and infiltration of the skin is usually present within face and hands (Figure 4). In half of sclerodermic patients with the limited form of the disease usually only the portion distal to the elbow joints is affected, but in the diffuse type the proximal areas are also involved—including the proximal parts of the limbs and the trunk. The advancement of skin lesions is evaluated usually using the modified Rodnan skin score [37]. Moreover, the ultrasound examination allows distinguish edematic lesions from fibrosis, but its reproducibility is relatively low [38]. High-frequency ultrasound offers a potential for objective and quantitative assessment of skin thickness and skin echogenicity in systemic sclerosis (SSc) [39]. It has been shown to be a valid measurement of skin thickening with excellent inter- and intraobserver variability (over 80% for interobserver variability and over 90% for intraobserver variability) [38,40]. High frequency ultrasound might be also useful in evaluation of oral mucosa, where fibrotic lesions are hyperechogenic [41].

Shear-wave elastography (SWE) was recently proven as a useful method of quantitative skin fibrosis assessment through the evaluation of skin strain. The elastic modulus values are significantly higher in SSc patients than in controls, with very accurate cutoff values, especially for hand fingers localization. Total scores of skin involvement determined at 17 sites (modified Rodnan skin thickness scores) correlate with skin stiffness SWE measurement [42].

SWE is more reproducible and has higher sensitivity than Rodnan Skin Score in the evaluation of skin condition in SSc, especially in case of changes non-detectable on physical evaluation, indicating it might become a useful tool in SSc diagnosis [43].

### 3.2. Musculoskeletal System

Tendon friction rub is a classical symptom, seen in over half of patients suffering from diffuse SSc, but only in 5% of patients with the limited form [44]. It is an early symptom, which might be seen even before skin lesions [45]. Ultrasound examination allows to visualise thickening of the tendon sheaths, which is a probable reason causing the physical symptom. Thickening of the A2 pulley was shown on ultrasound in a group of scleroderma patients to be greater compared to the control group [46]. Color Doppler ultrasound is able to prove an increased tissue blood flow which might be a sign of synovial proliferation due to articular inflammation [47].

### 3.3. Respiratory Tract

Ultrasound may be a useful accessory examination method in diagnosing pulmonary fibrosis, as a significant correlation between Kerley-B lines on ultrasound and interstitial opacifications on HRCT exam. Another potential marker for presence of fibrosis, visible on chest ultrasound are areas of irregular pleural thickening [48,49].

### 3.4. Cardiovascular System

Echocardiography remains the most frequently used modality in evaluation of PAH. PAH is present in approximately 10–15% of SSc patients [50]. The specificity of echocardiography is estimated at 75% and sensitivity at 90% [51]. Measuring of the peak velocity regurgitant velocity across the tricuspid valve is sufficient for preliminary assessment of PAH.

Echocardiography is also an effective method of evaluation of diastolic heart insufficiency in the course of cardiac fibrosis [52].

Right ventricular (RV) dysfunction can be primary (direct myocardial involvement) or secondary (due to PAH, LV dysfunction, or ILD). Echocardiographic RV imaging allows routine evaluation of the RV in SSc. Five parameters that are crucial in diagnosis of the RV dysfuncion include: RV fractional area change, tricuspid annular plane systolic excursion (TAPSE), RV free wall tissue Doppler S’, pulmonary artery systolic pressure, and the ratio of tricuspid regurgitant velocity to pulmonary artery velocity time integral (a noninvasive marker of pulmonary vascular resistance) [53].

The assessment of myocardial tissue velocities using the Doppler technique has provided immense insight into longitudinal function of the heart. Systolic longitudinal velocity can be used to identify early subclinical LV systolic dysfunction in patients with SSc whereas early diastolic tissue velocity is critically important as a marker of LV diastolic dysfunction [53].

During the last decade a number of reports regarding accelerated atherosclerosis in scleroderma patients. Elevated thickness of the intima-media complex in the common carotid artery has been proven among this group by various authors [54,55]. However, other investigators didn’t confirm those [56,57,58]. Other parameters that were evaluated included: velocity in the carotid and vertebral arteries, nitroglycerin or flow-mediated vasodilatation, and ankle-brachial index. Some of the publications confirmed earlier presence of those indirect symptoms of accelerated atherosis in SSc patients, while other exclude this correlation [59]. Regardless of those ambiguous results, when the abnormal lipid profile that was associated with scleroderma is taken into account [60], ultrasonographic examination is a valid and non-invasive tool in assessment of early atherotic lesions, which allows to begin treatment as soon as it is necessary.

### 3.5. Kidneys

Scleroderma renal crisis used to be a serious and often fatal complication of systemic sclerosis, but routine treatment with ACE-inhibitors significantly decreased its severity. Renal involvement is more common in the diffuse scleroderma. The standard method of evaluation and the renal function are functional examinations. Doppler assessment of the resistance index (RI) did not show correlation with a negative outcome [26].

## 4. Computed Tomography

### 4.1. Respiratory Tract

SSc-related interstitial lung disease (SSc-ILD) is the most common pulmonary manifestation of SSc. ILD and PH is the leading cause of mortality in patients suffering from scleroderma [61]. The basic method in diagnosing pulmonary symptoms of SSc is high resolution computed tomography (HRCT). Approximately two-third of patients present interstitial opacifications, even though the prevalence of complaints and symptoms found in physical examination is significantly lower. HRCT is a standard procedure in diagnosing and monitoring of treatment of the lung disease in SSc patients [62].

SSc-ILD shows numerous similarities to non-specific interstitial pneumonia (NSIP) both in medical imaging and histopathological examinations [63]. In HRCT (Figure 5) interstitial opacifications are seen in the dorso-basal parts of the lungs—usually of reticular morphology. Ground-glass opacities are also frequent, but typically they spare the subpleural parenchyma [64]. Symptoms that are not as characteristic of SSc but are reported in some of the patients include traction bronchiectasis and honeycombing, although the latter is typical rather in usual interstitial pneumonia than in NSIP [65,66]. The specificity of HRCT exceeds plain radiography, especially in the early stage of pulmonary disease (91% vs 39%) [67]. Abnormalities documented by CT examination correlate with anomalous results of functional tests. Moreover patients that are still asymptomatic in the functional exams might already develop interstitial lesions that might be visualised on a CT scan [6]. Presence of lung disease in HRCT has an impact on the response to treatment and mortality [68].

Cryptogenic organising pneumonia is one of the less frequent pulmonary manifestations of SSc [69]. Ground glass opacities, might be also a complication of pharmacological treatment of scleroderma [70]. Isolated pleural effusion (not including patients with cardiovascular insufficiency) is not a common symptom—seen in approximately 7% of patients [71]. Another example of a rather uncommon complication is pneumonia secondary to aspiration due to esophageal dysmotility. It is also possible that occult aspirations might have an important role in the pathogenesis of the pulmonary interstitial disease in scleroderma patients. In case of an acute aspiration, the distribution of lesions is typical - perihilar, multifocal, mostly in the posterior segments [72].

Additionally up to 60% of patients demonstrate slight lymphadenopathy of hilar and mediastinal nodes [61].

Computed tomography angiography (CTA) of the chest is the basic diagnostic tool used in evaluation of pulmonary embolism (PE). Although the morbidity in SSc patients isn’t higher than in general population, excluding this condition is important in case of acutely deteriorating respiratory function. Aside from typical symptoms—such as enlargement of the pulmonary trunk and its branches—in patients with chronic PE it might demonstrate calcifications within thrombi, scarring and areas of decreased lucency of the lung parenchyma. Irregular, mosaic attenuation is reported both in primary PAH and in post-thrombotic hypertension [62,73]

Symptoms of PAH on computed tomography (CT) examination include extension of the pulmonary arteries, right atrium and ventricle, hypertrophy of the right ventricle and a reflux of the contrast-enhanced blood to the superior vena cava. The last of listed symptoms has a high sensitivity for presence of PAH (of approximately 81%). Another practical indicator is the relation of the diameter of the pulmonary trunk and ascending aorta. A ratio exceeding 1 has high sensitivity and specificity (of 70% and 92% respectively) in diagnosing PH [62,74].

### 4.2. Gastrointestinal Tract

A standard chest CT performed to assess the lung parenchyma additionally allows to visualise the esophageal involvement in some of the patients. Esophagus filled with air, which is wider than 10 mm (Figure 6), or esophagus filled with fluid, is an equivalent of esophageal dismotility [75].

Ectasia of the antral part of the stomach (‘watermelon stomach’) is a less common finding, which might be seen in a contrast-enhanced CT of the abdominal cavity. Delayed emptying of the stomach might be also indirectly visualised, however both mentioned symptoms are better seen in an endoscopic examination [35].

Fibrosis in the small bowel might cause its dilation (>3 cm) and a higher number of mucosal folds (over 7 folds per inch) [76]. An asymmetrical fibrosis of one of the intestinal walls might lead to a buging of the contralateral wall (usually the antimesenteric wall). Dismotility of the small bowel is a possible secondary result of those lesions [77].

### 4.3. Heart

A CT examination of the chest allows to visualise pericardial effusion and pericardial fibrosis. Pericardial lesions are seen in a large amount of patients, but only 5–15% of them are symptomatic. Additionally, in some cases, pericardial effusion might be a symptom of cardiac insufficiency or PH, and not a symptom of scleroderma [78].

## 5. Magnetic Resonance Imaging

### 5.1. Skin

Pathologies involving skin and subcutaneous tissue are visible on magnetic resonance imaging (MRI) examinations in the majority of SSc patients [79].

### 5.2. Musculoskeletal System

MRI is a method that confirms pathologies in the musculoskeletal system even prior to patients’ complaints and positive physical examination [37]. A particular advantage of MRI examination is the ability to differentiate fibrosis (hypointense on T2-weighted images) and edema caused by inflammation (hyperintense on T2-weighted images) [18]. As many as 50% of patients demonstrated pathology within the articular structures of the hand (synovitis, tendinitis, erosions and bone marrow edema), but there is no correlation with the presence of clinical symptoms of arthritis [80]. Among patients with muscular symptoms, MRI confirmed findings of myositis, fascial and subcutaneous lesions in the vast majority of patients (78–89%), but to confirm this association [81].

Another reported affected site are the muscles of mastication. There is a correlation between fibrosis of the masseter muscle and presence of osteolytic lesions in the mandible [82].

### 5.3. Heart

MRI is a useful modality of evaluation the morphology of the heart as well as function and vitality of the cardiac muscle, specifically the right heart (Figure 7, Appendix A) [83], although comparison of MRI and the right heart catheterisation in assessment of PE was inconclusive. Bulging of the interventricular sept to the left correlates with the values of the pulmonary arterial pressure mostly in case of patients with advanced cases of PAH. Several studies have shown a delayed contrast enhancement of the right ventricle wall was seen in PAH patients (65–100%) [84,85]. While secondary to PAH right heart failure is the most severe cardiac complication in SSc patients, primary involvement of the heart is seen in a relatively large percent of patients—up to 15–35%. Cardiac fibrosis is the most common manifestation of primary heart involvement in scleroderma and might be seen in up to two-thirds of patients. Unlike other diagnostic imaging methods, MRI allows not only to visualise indirect symptoms of fibrosis (such as impaired ventricular filling and its enlargement) but also to directly asses the muscular involvement, eg. A delayed enhancement of the basal and central part of the left ventricle. Aside from fibrosis, some patients demonstrate also presence of myositis, which is seen as areas of increased signal intensity in T2-weighted images. Inflammation of the heart muscle is often seen in patients with symptoms of skeletal myositis [86,87].

The prevalence of ischaemic heart disease related to vasospasm is increased in SSc patients. The vasospasm is often triggered by low temperature, and this symptom is sometimes labeled as the Raynaud phenomenon of the heart. It might be confirmed by visualising subendocardial hypoperfusion areas during an MRI examination [88].

MRI ‘wall tagging’ and phase contrast techniques are useful to assess fibrosis and diastolic heart failure [89].

Native and post-contrast T1 mapping combined with T2 mapping and LGE (late gadolinium enhancement) discriminate between several myocardial diseases and notably between myocardial fibrosis and edema. The number of studies regarding T1 and T2 mapping is rapidly increasing and the available data show that native T1 mapping detects left ventricular elevated T1 in 50% of patients with SSc while a third of them had normal conventional screening (including standard CMR). This proves that T1 mapping cardiac MRI imaging frequently detects subclinical diffuse myocardial fibrosis in systemic sclerosis patients [90].

In patients without evidence of fibrosis (in LGE), T1 mapping parameters (extracellular volume fraction (ECV), gadolinium partition coefficient (λ), pre-contrast T1, and post-contrast T1) correlated with the modified Rodnan skin score, indicating a correlation between SSc cardiac and skin fibrosis. [91].

Moreover, T1-mapping offers incremental diagnostic value in the identification of patients with high risk of cardiovascular incidents and in addition to independently predict adverse outcomes at follow-up. Thus, CMR may be considered in cases where echocardiographic results are equivocal but the clinical picture warrants a higher index of suspicion, in order to identify high-risk patients early on and to institute prompt changes in the therapeutic management of SSc-related cardiac involvement [92].

### 5.4. Gastro-Intestinal Tract

MRI allows to recognize one of the more common manifestations of the SSc in the GI tract—lesions of the anus and rectum. Characteristic symptoms include ventral displacement of the anterior rectal wall, decreased thickness of the anal sphincter, and its reduced contrast enhancement caused by its fibrosis [93]. In the case of high intensity of the aforementioned lesions, patients might complain of stool inconsistency.

### 5.5. Lungs

A recent survey implies possible applications of MRI ultrafast echo-spin sequences in diagnosing interstitial pulmonary involvement—a correlation between the results of MRI and HRCT was proven [94].

## 6. Conclusions

In conclusion, SSc is associated with various radiological findings of specific organ and system manifestations of SSc, which may be apparent in its initial presentation and/or subsequent disease phases. This review may help clinicians and radiologists in the initial diagnosis and monitoring of progression of SSc.

## Figures and Tables

**Figure 1 jcm-10-00248-f001:**
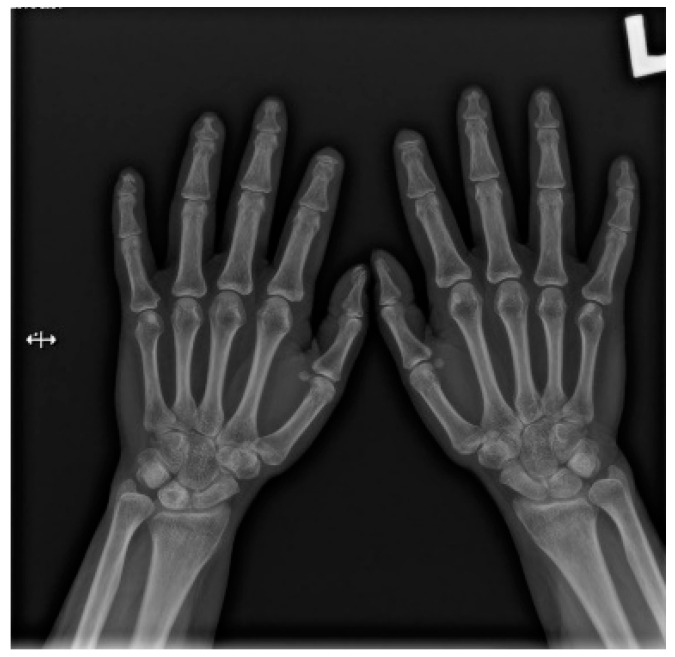
A-P view of the hand radiograph in a patient with systemic sclerosis shows acro-osteolysis especially expressed in the distal phalanx of the second finger of the left hand and in the distal phalanx of the second finger of the right hand.

**Figure 2 jcm-10-00248-f002:**
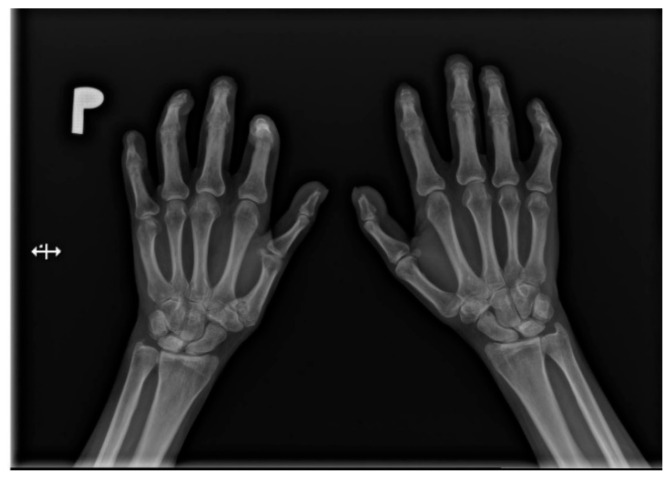
A-P view of the hand radiograph in a patient with systemic sclerosis shows acro-osteolysis of the distal phalanx of the first, second, third, and fourth fingers of both hands. Additionally, small joint contractures and areas of calcinosis in the fingertips are visible.

**Figure 3 jcm-10-00248-f003:**
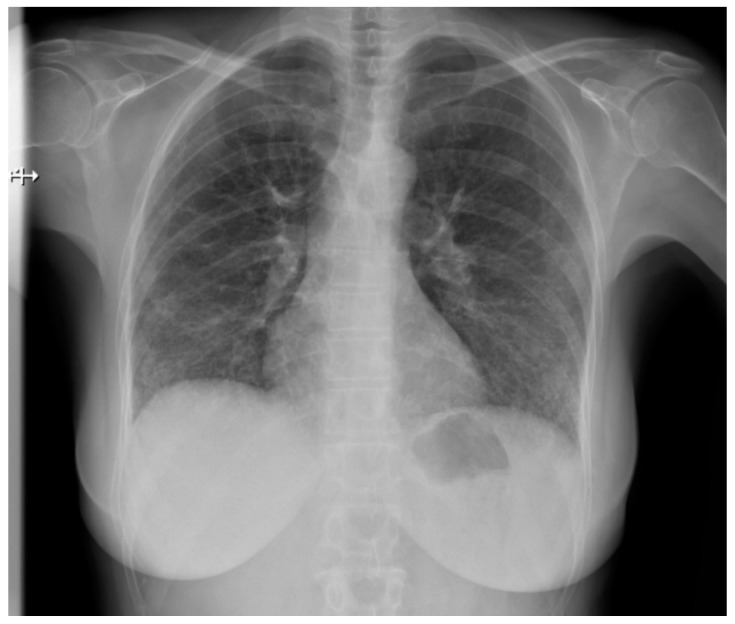
P-A view of the chest radiograph in a patient with systemic sclerosis demonstrates moderate pulmonary fibrosis.

**Figure 4 jcm-10-00248-f004:**
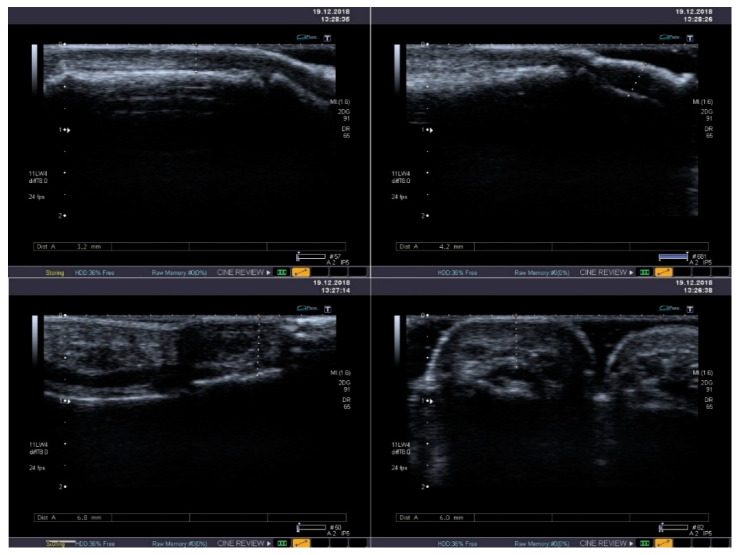
Standard B-mode ultrasound reveals mild thickening and heterogeneous echogenicity of soft tissues with its fibrosis in the thumb (**upper** panel) and in the second finger (**lower** panel).

**Figure 5 jcm-10-00248-f005:**
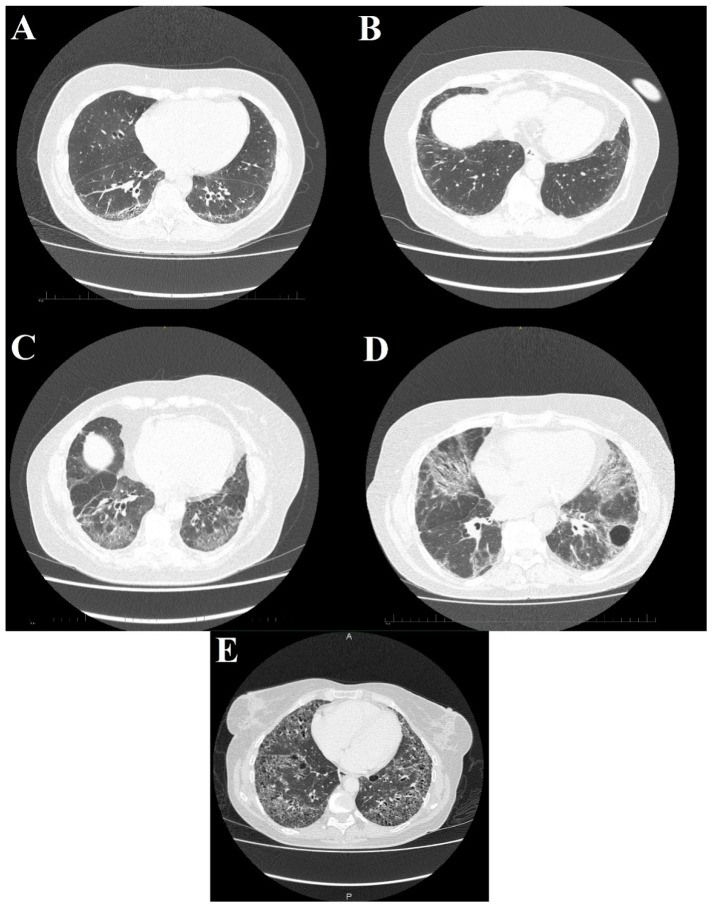
High-resolution computed tomography scans show different stages of systemic sclerosis-related interstitial lung disease. (**A**) Reticular, linear, and minor ground glass opacities; (**B**) ground glass opacities peripherally in the basal parts of the lower lobes; (**C**) diffuse ground glass, reticular opacities with discreet interlobular septal thickening in the basal parts of the lower lobes; (**D**) ground glass opacifications with reticular and linear patterns with traction bronchiectasis with an additional bulla; and (**E**) interstitial opacifications with extensive honeycombing.

**Figure 6 jcm-10-00248-f006:**
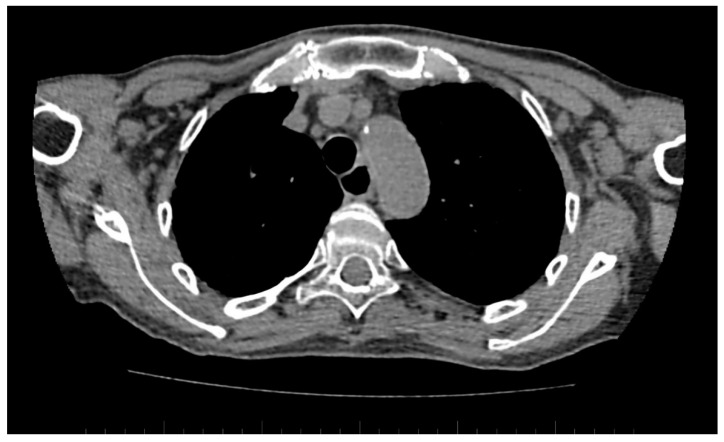
Computed tomography (CT) scan reveals dilated esophagus—RL diameter of approximately 20 mm.

**Figure 7 jcm-10-00248-f007:**
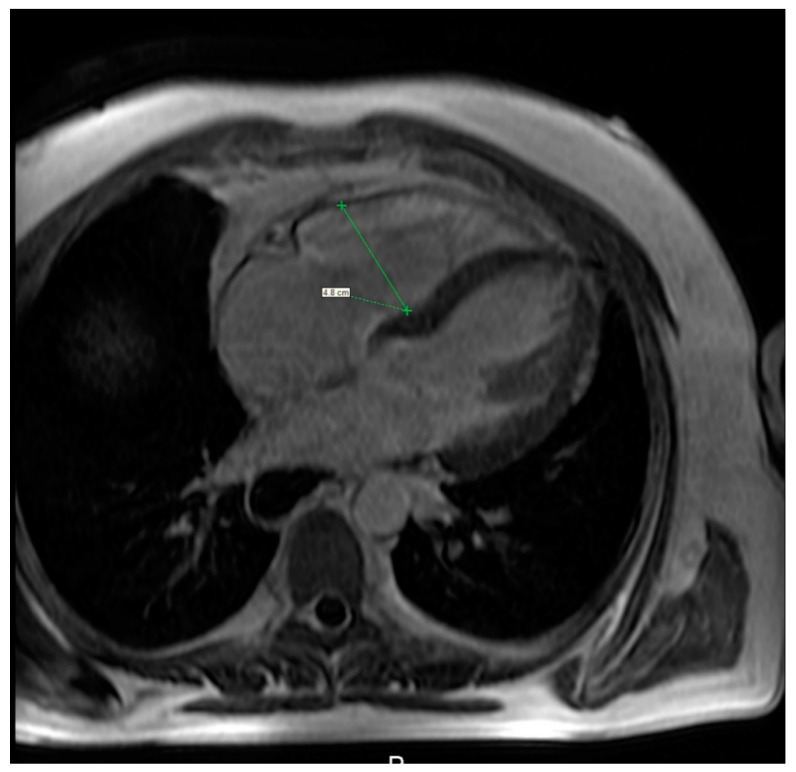
Cardiac magnetic resonance imaging of a 36-year-old woman with systemic sclerosis shows right ventricle dilation.

## Data Availability

The data presented in this study are available in references.

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
