# Peer review of "Imaging in Diagnosis of Systemic Sclerosis"

_jcm, 2021, doi:10.3390/jcm10020248_

Round 1
Reviewer 1 Report
I read with interest the paper by Rutka and colleagues about the role of imaging in SSc. The titles says diagnosis and treatment but it actually focuses only on diagnosis. No significant mention about SSc treatment is done in the text. The paper needs a full English revision and it is not adequately updated on the imaging modalities that have been implemented for SSc. There is an overall imprecision is the terms used and some specific topic are discussed only marginally. These are my comments:
- No SSc diagnostic criteria for exist. They are classificaiton criteria. ACR 1980 are not guidelines.
- Skin thickening proximal to the metacarpophalangeal joints is not a sufficient proof. As they are classification criteria is means patients with proximal skin fibrosis can be classified with SSc
- There is a major confusion about the term symptoms. Most of the times the authors refer to signs or manifestations.
- ACR/EULAR classification criteria include other domains which are NOT all symptoms (antoantibodies re not symptoms, so capillaroscopy and so on)
- Major confusion about PH and PAH. They are two completely different things.
- Vasoreactivity is not required either recommended in SSc patients with suspected PAH, it is for patients who belong to group 1 but not with CTD-associated PAH
- A iodine or barium swallow should be evaluaed at least every year in every SSc patient. This is not true or in line with EULAR EUSTAR recommendations
- Stomach is not a common location of lesions in scleroderma. This is not true. Watermelon stomach or GAVE is a complication of SSc
- Echocardiography is also used for heart involvement in SSc (myocarditis).
- CMR paragraph should be implemented. New methodologies are used to properly assess heart involvement including T1 and T2 mapping.
Author Response
Dear Editor,
Thank you for giving us the opportunity to submit a revised version of the manuscript “Imaging in diagnosis of systemic sclerosis” to reconsider it for publication in the Journal of Clinical Medicine. We appreciate the time and effort that you and the reviewers dedicated to providing feedback on our manuscript and are grateful for the insightful comments on and valuable improvements to this paper. We have incorporated the majority recommendations made by the reviewers. Those changes are highlighted with red color within the manuscript. Please see below, in red, for a point-by-point response to the reviewers’ comments and concerns.
- No SSc diagnostic criteria for exist. They are classificaiton criteria. ACR 1980 are not guidelines. Skin thickening proximal to the metacarpophalangeal joints is not a sufficient proof. As they are classification criteria is means patients with proximal skin fibrosis can be classified with SSc.
It was corrected.
- There is a major confusion about the term symptoms. Most of the times the authors refer to signs or manifestations.
This is true, so we have modified the title of the manuscript
- Major confusion about PH and PAH. They are two completely different things.
It was corrected.
- Vasoreactivity is not required either recommended in SSc patients with suspected PAH, it is for patients who belong to group 1 but not with CTD-associated PAH
It was corrected.
- A iodine or barium swallow should be evaluaed at least every year in every SSc patient. This is not true or in line with EULAR EUSTAR recommendations.
It was corrected.
- Stomach is not a common location of lesions in scleroderma. This is not true. Watermelon stomach or GAVE is a complication of SSc
It was corrected.
- Echocardiography is also used for heart involvement in SSc (myocarditis).
It was added to the article.
- CMR paragraph should be implemented. New methodologies are used to properly assess heart involvement including T1 and T2 mapping.
It was added to the article.
Reviewer 2 Report
This could be an interesting review but the way of presenting the data to my view limit its interest and the usefulness for the reader.
Major comments
I would suggest to present the data in a different way: instead of classifying according to imaging methods, i think it would be clearer for the reader to present each involvement and then the differents imaging methods to detect/assess this involvement.
In addition, within musculoskeletal system, I would suggest to divide the involvements in Synovial, Muscle, Tendon, Soft-tissue (calcinosis) and describe the methods to assess each of these involvements.
Consistently for heart involvement, presenting the imaging methods to assess separately myocarditis, diastolic dysfuntion, PAH would add the value of the meassage, being clearer and allowing the clinician in practice to know which imaging is needed in which situation.
A table with (1) indications of this imaging (e.g. diagnose, staging, treatment response) and of (2) pitfalls and advantages of each imaging method for each involvement would be good way to summarize the results.
Moreover, the method to obtain the review is not described in the text: systematic review on pubmed, which terms employed, guidelines...
Other comments: Please correct in the text
- stomach involvement is common in SSc, unlike the statement of the authors (Rheum Dis Clin North Am 2018,doi: 10.1016/j.rdc.2017.09.002)
- The authors should consider the elastography as an imaging method for skin assessment
- PAH is more present in limited forms and not diffuse, unlike the author's statement
- For renal crisis, the treatment allowed to decrease the severity but not the prevalence (ACE inhibitors are not used as preventive treatment)
Author Response
Dear Editor,
Thank you for giving us the opportunity to submit a revised version of the manuscript “Imaging in diagnosis of systemic sclerosis” to reconsider it for publication in the Journal of Clinical Medicine. We appreciate the time and effort that you and the reviewers dedicated to providing feedback on our manuscript and are grateful for the insightful comments on and valuable improvements to this paper. We have incorporated the majority recommendations made by the reviewers. Those changes are highlighted with red color within the manuscript. Please see below, in red, for a point-by-point response to the reviewers’ comments and concerns.
1. I would suggest to present the data in a different way: instead of classifying according to imaging methods, i think it would be clearer for the reader to present each involvement and then the differents imaging methods to detect/assess this involvement. In addition, within musculoskeletal system, I would suggest to divide the involvements in Synovial, Muscle, Tendon, Soft-tissue (calcinosis) and describe the methods to assess each of these involvements. Consistently for heart involvement, presenting the imaging methods to assess separately myocarditis, diastolic dysfuntion, PAH would add the value of the meassage, being clearer and allowing the clinician in practice to know which imaging is needed in which situation.
This is a very good idea, but it would be too similar to some of the articles we have read.
2. A table with (1) indications of this imaging (e.g. diagnose, staging, treatment response) and of (2) pitfalls and advantages of each imaging method for each involvement would be good way to summarize the results.
The table proposal was added.
3. Moreover, the method to obtain the review is not described in the text: systematic review on pubmed, which terms employed, guidelines...
It was added
4. Other comments: Please correct in the text
- stomach involvement is common in SSc, unlike the statement of the authors (Rheum Dis Clin North Am 2018,doi: 10.1016/j.rdc.2017.09.002)
It was corrected
- The authors should consider the elastography as an imaging method for skin assessment
It was added to the article
- PAH is more present in limited forms and not diffuse, unlike the author's statement
It was corrected
- For renal crisis, the treatment allowed to decrease the severity but not the prevalence (ACE inhibitors are not used as preventive treatment)
It was corrected.
Round 2
Reviewer 2 Report
No further comment